# AUTOMATIC TERMINATION FOR HYPERPARAMETER OPTIMIZATION

## ABSTRACT

Bayesian optimization (BO) is a widely popular approach for the hyperparameter optimization (HPO) of machine learning algorithms. At its core, BO iteratively evaluates promising configurations until a user-defined budget, such as wall-clock time or number of iterations, is exhausted. While the final performance after tuning heavily depends on the provided budget, it is hard to pre-specify an optimal value in advance. In this work, we propose an effective and intuitive termination criterion for BO that automatically stops the procedure if it is sufficiently close to the global optima. Across an extensive range of real-world HPO problems, we show that our termination criterion achieves better test performance compared to existing baselines from the literature, such as stopping when the probability of improvement drops below a fixed threshold. We also provide evidence that these baselines are, compared to our method, highly sensitive to the choices of their own hyperparameters. Additionally, we find that overfitting might occur in the context of HPO, which is arguably an overlooked problem in the literature, and show that our termination criterion mitigates this phenomenon on both small and large datasets.

## 1 INTRODUCTION

While the performance of machine learning algorithms crucially depends on their hyperparameters, setting them correctly is typically a tedious and expensive task. Hyperparameter optimization (HPO) emerged as a new sub-field in machine learning that tries to automatically determine how to configure a machine learning algorithm. One of the most successful strategies for HPO is Bayesian optimization (BO; Močkus, 1975; Chen et al., 2018; Snoek et al., 2012; Melis et al., 2018) which iteratively trains a probabilistic model on the evaluations of the tuned algorithm. to select the most promising next candidate point that trades-off exploration and exploitation.

In practice, the quality of the final solution found by BO heavily depends on a user defined budget, such as wall-clock time or the number of iterations, which needs to be defined in advanced. If this budget is too small, BO might return hyperparameters that result in a poor predictive performance. If the budget is too large, compute resources will be wasted and, in some cases, it may result in *overfitting* as we will show in our experiments. Automatically stopping BO is a rather under-explored topic in the literature. A simple baseline is to stop if BO has not found a better solution than the current best incumbent for some successive iterations, which is in the same vein as early stopping for neural network training. Another approach is to track probability of improvement (Lorenz et al., 2016) or expected improvement (Nguyen et al., 2017), and stop the optimization process once it falls below a given threshold. However, determining this threshold may in practice be less intuitive than setting the number of iterations or the wall-clock time. Instead of stopping BO completely, McLeod et al. (2018) propose to switch to local optimization when the global regret is smaller than a pre-defined target. This condition can also be used to terminate BO early, but it comes with additional complexity such as identifying a (convex) region for local optimization and again a predefined budget.

In this work, we propose a simple and interpretable automatic termination criterion for BO. In our criterion, we construct high-probability confidence bound on the regret (i.e., the difference of our current solution to the global optimum) exploiting the probabilistic model of the objective. Thus, users are now asked to specify a desired tolerance that defines *how accurate should the final solution be compared to global optimum*. In addition, we propose to determine the threshold via a cross-validation estimate of the generalization error. This choice takes into account the irreducible discrepancy between

the actual objective and the target function optimized via BO, namely the difference between the performance on new data (i.e., the population risk) and the validation error. Our extensive empirical evaluation on a variety of HPO and NAS benchmarks suggests that our method is more robust and effective in maintaining the final solution quality than common baselines (Lorenz et al., 2016; Nguyen et al., 2017). We also surface overfitting effects in HPO on both small and large datasets, arguably an overlooked problem in the literature, and demonstrate that our termination criterion helps to mitigate it.

## 2 BACKGROUND

**Bayesian optimization** (BO) refers to methods of optimizing a black-box objective $f : \Gamma \to \mathbb{R}$ in an iterative manner. At every step $t$, a learner selects an input $\gamma_t \in \Gamma$ and observes a noisy output

$$y_t \triangleq f(\gamma_t) + \varepsilon_t,$$

where $\varepsilon_t$ is typically assumed to be i.i.d. (sub)-Gaussian noise with some variance (proxy) $\sigma_\varepsilon^2$. The decision of the next input to evaluate depends on a probabilistic model, used to approximate the objective $f$, and an acquisition function, which determines the decision rule. A popular choice for the probabilistic model is a *Gaussian process (GP)*: $f \sim GP(\mu, \kappa)$ (Rasmussen & Williams, 2006), specified by some mean function $\mu : \Gamma \to \mathbb{R}$ and some kernel $\kappa : \Gamma \times \Gamma \to \mathbb{R}$. As observations $y_{1:t} = [y_1, \ldots, y_t]^\top$ for the selected inputs $G_t = \{\gamma_1, \ldots, \gamma_t\}$ are being collected, they are used to update the posterior belief of the model defined by the posterior mean $\mu_t(\gamma)$ and variance $\sigma_t^2(\gamma)$ as:

$$\mu_t(\gamma) = \kappa_t(\gamma)^T (K_t + \sigma_\varepsilon^2 I)^{-1} y_{1:t} \tag{1}$$

$$\sigma_t^2(\gamma) = \kappa(\gamma, \gamma) - \kappa_t(\gamma)^\top (K_t + \sigma_\varepsilon^2 I)^{-1} \kappa_t(\gamma), \tag{2}$$

where $(K_t)_{i,j} = \kappa(\gamma_i, \gamma_j)$ and $\kappa_t(\gamma)^T = [\kappa(\gamma_1, \gamma), \ldots, \kappa(\gamma_t, \gamma)]^T$. The next input to query is determined by an *acquisition function* that aims to trade off exploration and exploitation. Common choices include probability of improvement (Kushner, 1963), entropy search (Hennig & Schuler, 2012), GP upper-confidence bound (Srinivas et al., 2010), to name a few.

The convergence of BO can be quantified via *(simple) regret*, i.e., the sub-optimality in function value:

$$r_t := f(\gamma_t^*) - f(\gamma^*), \tag{3}$$

where $\gamma^*$ is the global optimizer of $f$ and $\gamma_t^* = \arg\min_{\gamma \in G_t} f(\gamma)$. Specifying adequate tolerance that defines how small the regret should be to terminate BO is of high importance as it determine both the quality and the cost of the solution. However, this criterion cannot be directly evaluated in practice, as the input $\gamma^*$ and the optimum $f(\gamma^*)$ are not known.

**Hyperparameter optimization** (HPO) is a widely considered application for BO. Consider a supervised learning setting training a machine learning model (e.g., a neural network) $\mathcal{M}$ on some feature-response data points $\mathcal{D} = \{(x_i, y_i)\}_{i=1}^n$ sampled i.i.d. from some unknown data distribution $P$. The model is obtained by running a training algorithm (e.g., optimizing the weights of the neural network via SGD) on $\mathcal{D}$, and the model returned also depends on *hyperparameters* $\gamma$ (e.g., learning rates used, batch size, etc.). We use the notation $\mathcal{M}_\gamma(x; \mathcal{D})$ to refer to the prediction that the model produced by $\mathcal{M}$ makes for an input x, when trained with hyperparameters $\gamma$ on data $\mathcal{D}$. Given some loss function $\ell(\cdot, \cdot)$, the *population risk* of the model on unseen data points is given by the expected loss $\mathbb{E}_P[\ell(y, \mathcal{M}_\gamma(x, \mathcal{D}))]$. The main objective of HPO is to identify hyperparameters $\gamma$, such that the resulting model minimizes the population risk:

$$f(\gamma) = \mathbb{E}_P\big[\ell\big(y, \mathcal{M}_\gamma(x, \mathcal{D})\big)\big], \qquad \gamma^* = \arg\min_{\gamma \in \Gamma} f(\gamma). \tag{4}$$

In practice, however, the population risk cannot be evaluated since $P$ is unknown. Thus, typically, it is estimated on a separate finite validation set $\mathcal{D}_V$ drawn from the same distribution $P$. As the result, practical HPO focuses on minimizing the *empirical estimate* $\hat{f}(\gamma)$ of the expected loss $f(\gamma)$ leading to (probably different) optimizer $\gamma_\mathcal{D}^*$:

$$\hat{f}(\gamma) = \frac{1}{|\mathcal{D}_V|} \sum_{x_i, y_i \in \mathcal{D}_V} \ell\big(y_i, \mathcal{M}_\gamma(x_i, \mathcal{D})\big), \qquad \gamma_\mathcal{D}^* = \arg\min_{\gamma \in \Gamma} \hat{f}(\gamma). \tag{5}$$

At its core, BO-based HPO evaluates noisy empirical estimate $\hat{f}(\gamma_t)$ for promising hyperparameters $\gamma_t$ for some finite number of iterations and the final performance after termination heavily depends on that number. Alternatively, one can also terminate BO when sufficiently close to the global optima, i.e., using an analogue to the simple regret $r_t$ for the validation loss $\hat{f}(\gamma)$ and $\hat{f}(\gamma_t^*) = \min_{\gamma \in G_t} \hat{f}(\gamma)$:

$$\hat{r}_t := \hat{f}(\gamma_t^*) - \hat{f}(\gamma_\mathcal{D}^*). \tag{6}$$

**Inconsistency in the optimization objective.** Importantly, the true HPO objective $f(\gamma)$ in Eq. (4) and the empirical surrogate $\hat{f}(\gamma)$ in Eq. (5) used for tuning by BO generally do not coincide. Therefore, existing BO approaches may yield sub-optimal solutions to the population risk minimization, even if they succeed in globally optimizing $\hat{f}(\gamma)$. This issue, however, is typically neglected in practical HPO. In contrast, we propose a termination condition for BO motivated by the discrepancy in the objectives.

## 3 TERMINATION CRITERION FOR HYPERPARAMETER OPTIMIZATION

This section firstly motivates why early termination of HPO can be beneficial and then addresses the following two questions: (1) How to estimate the unknown simple regret and (2) What threshold of the simple regret can be used to stop HPO.

### 3.1 MOTIVATION FOR THE TERMINATION CRITERION

We start by analysing the true discrepancy of interest: between the population risk $f(\gamma_t^*)$ at some input $\gamma_t^*$ and true optimum $f(\gamma^*)$. We then observe that this discrepancy sums from the statistical error of the empirical BO objective $\hat{f}(\gamma)$ as well the sub-optimality of the BO candidates (encoded in the simple regret $\hat{r}_t$). The key insight of the following proposition is that iterative reducing of $\hat{r}_t$ to 0 may not bring any benefits if the statistical error dominates.

**Proposition 1.** *Consider the expected loss $f$ and its estimator $\hat{f}$ defined in Eqs. (4) and (5), respectively, and assume the statistical error of the estimator is bounded as $||\hat{f} - f||_\infty \leq \epsilon_{st}$ for some $\epsilon_{st} \geq 0$. Let $\gamma^*$ and $\gamma_\mathcal{D}^*$ be their optimizers: $\gamma^* = \arg\min_{\gamma \in \Gamma} f(\gamma)$ and $\gamma_\mathcal{D}^* = \arg\min_{\gamma \in \Gamma} \hat{f}(\gamma)$. Let $\gamma_t^*$ be some candidate solution to $\min_{\gamma \in \Gamma} \hat{f}(\gamma)$ with sub-optimality in function value $\hat{r}_t := \hat{f}(\gamma_t^*) - \hat{f}(\gamma_\mathcal{D}^*)$. Then the gap in generalization performance $f(\gamma_t^*) - f(\gamma^*)$ can be bounded as follows:*

$$f(\gamma_t^*) - f(\gamma^*) \leq \underbrace{f(\gamma_t^*) - \hat{f}(\gamma_t^*)}_{\leq \epsilon_{st}} + \underbrace{\hat{f}(\gamma_t^*) - \hat{f}(\gamma_\mathcal{D}^*)}_{=\hat{r}_t} + \underbrace{\hat{f}(\gamma_\mathcal{D}^*) - \hat{f}(\gamma^*)}_{\leq 0} + \underbrace{\hat{f}(\gamma^*) - f(\gamma^*)}_{\leq \epsilon_{st}}$$

$$\leq 2\epsilon_{st} + \hat{r}_t.$$

*Moreover, without further restrictions on $f$, $\hat{f}$, $\gamma_t^*$ and $\gamma^*$, the upper bound is tight.*

*Proof:* While the second inequality is due to the definition of $\gamma_t^*$, the others can proved as follows:

$$f(\gamma_t^*) - \hat{f}(\gamma_t^*) \leq |f(\gamma_t^*) - \hat{f}(\gamma_t^*)| \leq \max_{\gamma \in \Gamma} |f(\gamma) - \hat{f}(\gamma)| = ||\hat{f} - f||_\infty \leq \epsilon_{st},$$

$$\gamma_\mathcal{D}^* = \arg\min_{\gamma \in \Gamma} \hat{f}(\gamma) \longrightarrow \forall \gamma \in \Gamma : \hat{f}(\gamma_\mathcal{D}^*) - \hat{f}(\gamma) \leq 0 \longrightarrow \hat{f}(\gamma_\mathcal{D}^*) - \hat{f}(\gamma^*) \leq 0. \quad \blacksquare$$

The proposition provides the discrepancy bound in terms of the statistical error $\epsilon_{st}$ and simple regret $\hat{r}_t$. This naturally incites terminating HPO at a candidate $\gamma_t^*$ for which the simple regret $\hat{r}_t$ is of the same magnitude as the statistical error $\epsilon_{st}$ (as further reduction in $\hat{r}_t$ may not improve notably the true objective). However, neither of the quantities $\epsilon_{st}$ and $\hat{r}_t$ is known.

Below, we propose a termination criterion that relies on estimates of both quantities. Firstly, we show how to use confidence bounds on $\hat{f}(\gamma)$ to obtain high probability upper bounds on the simple regret $\hat{r}_t$ Srinivas et al. (2010); Ha et al. (2019). Secondly, we estimate the statistical error $\epsilon_{st}$ in the case of cross-validation (Stone, 1974; Geisser, 1975) where the model performance is defined as an average over several training-validation runs. To this end, we rely on the statistical characteristics (i.e., variance or bias) of such cross-validation-based estimator that are theoretically studied by Nadeau & Bengio (2003); Bayle et al. (2020). When cross validation is not used, one could define an intuitive threshold in advance due to the usage of interpretable upper bound of simple regret.

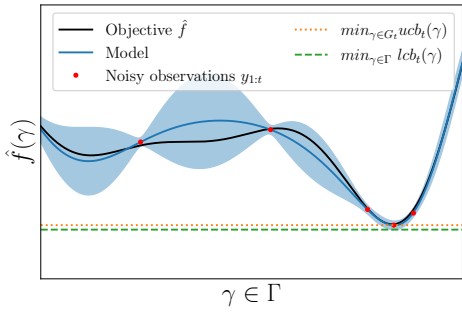
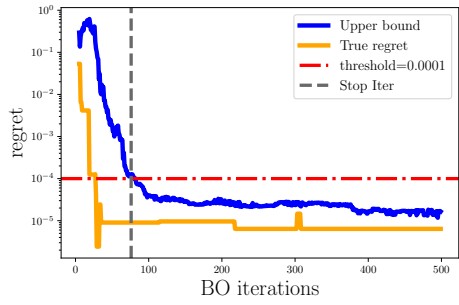

(a) Key ingredients of the termination criterion  (b) Example of triggering the termination

Figure 1: (a) Visualization of the upper bound for $\hat{r}_t$. The gap between green and orange lines is the estimate of the upper bound for $\hat{r}_t$ (b) Illustration of automated BO termination when tuning MLP on the `naval` dataset from HPO-Bench (Klein & Hutter, 2019) with the BORE optimizer (Tiao et al., 2021).

## 3.2 BUILDING BLOCKS OF THE TERMINATION CRITERION

**Upper bound for the simple regret $\hat{r}_t$.** The key idea behind bounding $\hat{r}_t$ is that, as long as the GP-based surrogate model of $\hat{f}(\cdot)$ is well-calibrated, we can use it to construct high-probability confidence bounds for $\hat{f}(\cdot)$. In particular, Srinivas et al. (2010) show that as long as $\hat{f}$ has bounded norm in the reproducing kernel Hilbert space (RKHS) associated with the covariance function $\kappa$ used in the GP, $\hat{f}(\gamma)$ is bounded (with high probability) by lower and upper confidence bounds $\mathrm{lcb}_t(\gamma) = \mu_t(\gamma) - \sqrt{\beta_t}\sigma_t(\gamma)$ and $\mathrm{ucb}_t(\gamma) = \mu_t(\gamma) + \sqrt{\beta_t}\sigma_t(\gamma)$. Hereby, $\beta_t$ is a parameter that ensures validity of the confidence bounds (see Appendix A.2.3 for practical discussion and ablation study).

Consequently, we can bound the unknown $\hat{f}(\gamma_t^*)$ and $\hat{f}(\gamma_{\mathcal{D}}^*)$ that define the sub-optimality $\hat{r}_t$:

$$\hat{r}_t = \hat{f}(\gamma_t^*) - \hat{f}(\gamma_{\mathcal{D}}^*) \leq \min_{\gamma \in G_t} \mathrm{ucb}_t(\gamma) - \min_{\gamma \in \Gamma} \mathrm{lcb}_t(\gamma) =: \bar{r}_t, \tag{7}$$

where the inequality for $\hat{f}(\gamma_t^*)$ is due to the definition of the reporting rule $\hat{f}(\gamma_t^*) = \min_{\gamma \in G_t} \hat{f}(\gamma)$ over the evaluated points $G_t = \{\gamma_1, \ldots, \gamma_t\}$. We illustrate the idea with an example in Fig. 1a.

**Termination threshold.** We showed how to control the optimization error via the (computable) regret upper bound $\bar{r}_t$ in the above and now we propose to stop the BO when $\bar{r}_t$ is smaller than some threshold $\epsilon_{BO}$, i.e. at an iteration $T : \bar{r}_T \leq \epsilon_{BO}$. Following Proposition 1, we suggest setting $\epsilon_{BO}$ to be of similar magnitude as the statistical error $\epsilon_{st}$ (since smaller regret $\hat{r}_t$ is not beneficial when $\epsilon_{st}$ dominates). We now discuss how to estimate the statistical error $\epsilon_{st}$ in case of cross-validation.

Cross-validation is the standard approach to compute an estimator $\hat{f}(\gamma)$ of the population risk. The data $\mathcal{D}$ is partitioned into $k$ equal-sized sets $\mathcal{D}_1, \ldots, \mathcal{D}_k$ used for (a) training the model $\mathcal{M}_\gamma(\cdot; \mathcal{D}_{-i})$, where $\mathcal{D}_{-i} = \cup_{j \neq i} \mathcal{D}_j$ (i.e., training on all but the $i$-th fold), and (b) validating $\mathcal{M}_\gamma(\cdot; \mathcal{D}_i)$ on the $i$-th fold of the data. These two steps are then repeated in a loop $k$ times, and then the average over $k$ validation results is computed, yielding the noisy BO evaluation $y = \hat{f}(\gamma) + \varepsilon$ where the noise is due to the randomness in the training procedure or to epistemic uncertainty.

The statistical error $\epsilon_{st}$ of an estimate can be characterised in terms of its variance and bias where the latter can be neglected in case of cross-validation (Bayle et al., 2020). Though the variance of the cross-validation estimate $\mathrm{Var}\hat{f}(\gamma) = \mathbb{E}[(\hat{f}(\gamma) - \mathbb{E}\hat{f}(\gamma))^2]$ is generally unknown, Nadeau & Bengio (2003) propose an estimate for it. Specifically, for the sample variance (denoted as $s_{\mathrm{cv}}^2$) of $k$-fold cross-validation, a simple post-correction technique to estimate the variance $\mathrm{Var}\hat{f}(\gamma)$ is

$$\mathrm{Var}\hat{f}(\gamma) \approx \left( \frac{1}{k} + \frac{|\mathcal{D}_i|}{|\mathcal{D}_{-i}|} \right) s_{\mathrm{cv}}^2(\gamma), \tag{8}$$

where $|\mathcal{D}_i|, |\mathcal{D}_{-i}|$ are the set sizes. For example, in the case of 10-fold cross-validation we have $\mathrm{Var}\hat{f}(\gamma) \approx 0.21 s_{\mathrm{cv}}^2(\gamma)$. We are now ready to propose our termination condition in the following.

**Termination condition for BO.** *Consider the setup of Proposition 1 where $\hat{f}(\cdot)$ is a cross-validation-based estimator being iteratively minimized by BO. Let $y_t = \hat{f}(\gamma_t) + \varepsilon_t$ be a noisy evaluation at the input $\gamma_t$ and $\gamma_t^* = \underset{\gamma_i \in \{\gamma_1, \ldots, \gamma_t\}}{\arg\min} \, y_i$ denote the best observed input by BO iteration $t$. Let $\bar{r}_t$ defined in Eq. (7) be the simple regret bound computed at each iteration $t$. Let the variance $\operatorname{Var}\hat{f}(\gamma_t^*)$ of the estimator $\hat{f}(\cdot)$ be approximated according to Eq. (8). Then, BO is terminated once:*

$$\bar{r}_t < \sqrt{\operatorname{Var}\hat{f}(\gamma_t^*)}. \tag{9}$$

Intuitively, the termination is triggered once the maximum plausible improvement becomes less than the standard deviation of the estimate. This variance-based termination condition adapts to different algorithms or datasets and its computation comes with negligible computational cost on top of cross-validation. The pseudo code for the criterion is summarised in Appendix A.1.2. If cross-validation cannot be used or is computationally prohibitive, the user can define the right-hand side of the termination condition. In this case, the upper bound on the left-hand side still has an intuitive interpretation: the user can set the threshold based on their desired solution accuracy. This case is demonstrated in Fig. 1b, with an example of automatic termination for tuning an MLP.

## 4 EXPERIMENTS

The main challenge of early stopping HPO is to find the right trade-off between reducing runtime and performance degradation. We thus study in experiments how the speed-up gained from different termination criterions affects the final test performance.

### 4.1 EXPERIMENTAL SETUP

We start with describing our experimental setup to validation our approach. To establish a sensible experimental setup, we define two new metrics that account for the trade-off between resources saved and drop in final performance and provide a list of reasonable baselines.

#### 4.1.1 BASELINES

To validate our proposed termination criterion we consider the following baselines:

- The first baseline is a näive convergence test controlled by a parameter $i$: BO is stopped once the *best* observed validation metric remains unchanged for $i$ consecutive iterations. This convergence condition heavily relies on $i$, which needs to be chosen in advance. We consider values commonly used in practice, namely $i = \{10, 30, 50\}$.

- We also compare to Lorenz et al. (2016); Nguyen et al. (2017), which terminates BO once the value of the Probability of Improvement (PI) or Expected Improvement (EI) drops below a pre-defined threshold. Both acquisition functions do not live on a well-defined scale, rendering it hard to identify a sensible threshold in practice. We follow the recommendations from Nguyen et al. (2017); Lorenz et al. (2016) and consider the following values for EI $\{10^{-9}, 10^{-13}, 10^{-17}\}$ and for PI $\{10^{-5}, 10^{-9}, 10^{-13}\}$. Empirically, we observe that all thresholds lead to significant speeds up but, at the same time, to a severe degradation of the final solution quality. Because of space constraints, we only show the results for $10^{-17}$ for EI and $10^{-13}$ for PI, and report the full results in Appendix A.2.

#### 4.1.2 METRICS

To measure the effectiveness of a termination criterion, we analyze two metrics to quantify the change in test error and the time saved. Particularly, given a BO budget T, we compare the test error $y_{es}$ when early stopping is triggered to the test error $y_T$. For each experiment, we compute the *relative test error change* RYC as

$$\text{RYC} = \frac{y_T - y_{es}}{\max(y_T, y_{es})}. \tag{10}$$

This allows us to aggregate the results over different algorithms and datasets, as RYC $\in [-1, 1]$ and can be interpreted as follows: A positive RYC represents an improvement in the test error when

applying early stopping, while a negative RYC indicates the opposite. Similarly, let the total training time for a predefined budget $T$ be $t_T$ and the total training time when early stopping is triggered be $t_{es}$. Then the *relative time change* RTC is defined as

$$\text{RTC} = \frac{t_T - t_{es}}{t_T}. \tag{11}$$

A positive RTC, where RTC $\in [0, 1]$, indicates a reduction in total training time. While reducing training time is desirable, it should be noted that this can be achieved through any simple stopping criterion (e.g., consider interrupting HPO with a fixed probability after every iteration). In other words, the RTC is not a meaningful metric when decoupled from the RYC and we will thus consider the two in tandem in the following experiments.

### 4.2 How to select the data to estimate the bound?

Since we are only interested in estimating the upper bound of the regret of the incumbent, we conjecture that using only the top performing hyperparameter evaluations may improve the estimation quality. To validate this, we use BORE (Tiao et al., 2021) tuning results on the `naval` dataset from HPO-Bench Klein & Hutter (2019) (described in Section 4.4) where we can quantify the true regret. We compute the upper bound by Eq. (7) using three options: 100%, top 50% or top 20% of the hyperparameters evaluated so far. The quality of the bound is measured by the difference to the true regret. Results are shown in Fig. 2a, where the median of 50 replicates are shown as solid line and the 20'th and 80'th quantiles are shown as dashed and dotted lines, respectively. We also show the number of negative differences (the upper bound is smaller than the true regret) in the legend next to the different options in the this figure.

From Fig. 2a, fitting a surrogate model with all the hyperparameter evaluations poses a challenge for estimating the upper bound of the regret, which is aligned with recent findings on more efficient BO with local probabilistic model, especially for high-dimensional problems (Eriksson et al., 2019). Using the top 20% evaluations gives the best upper bound estimation quality in the median, at the cost of the most under-estimations of the true regret (2553). Our method would stop too early due to the under-estimation, which would negatively impact the quality measured by RYC scores, as shown in Fig. 2b. *As a result, we use top 50% hyperparameters evaluations for the upper bound estimation throughout this paper.*

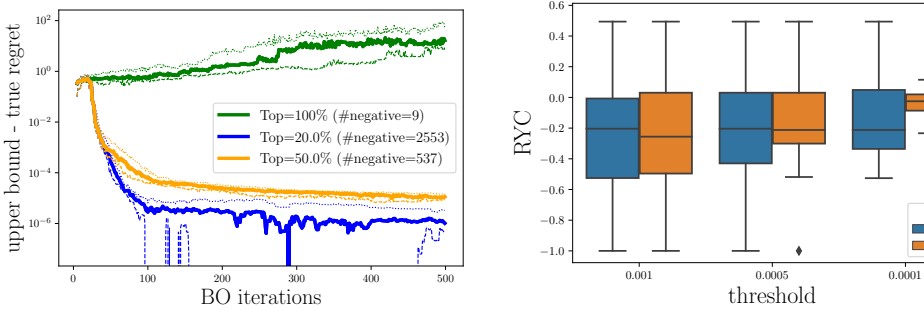

(a) Bound quality for using all, the top 50% and top 20% hyperparameter evaluations, measured by the difference between upper bound and true regret.

(b) Box plots of RYC scores when using the top 50% and top 20% hyperparameter evaluations under common thresholds.

Figure 2: The upper bound estimation quality is affected by the set of hyperparameters evaluations used in the surrogate model training.

### 4.3 Standard BO with cross-validation

We use standard BO with matérn 5/2 kernel in the GP surrogate. The hyperparameters of the GP are estimated with type II maximum likelihood estimation. We tune two algorithms: XGBoost (XGB) with 9 hyperparameters and Random Forest (RF) with three hyperparameters on 19 small tabular datasets using cross-validation. The detailed BO setting, hyperparameter search space of the

algorithms, as well as the characteristics of the datasets can be found in Appendix A.1. We optimize classification error or rooted mean square error for classification and regression datasets, respectively.

For our stopping method, we use Eq. (8) as our stopping threshold. We apply automatic termination only after the first 20 iterations to ensure a robust fit of the surrogate models both for our method and the baselines. We present the aggregated results across datasets for XGB and RF in Fig. 3 where the mean (shown in the dots) and standard deviation (shown in the error bars in both RTC and RYC dimensions) of RTC and RYC scores for different termination methods are shown.

From Fig. 3, a general trend on $i$ in the convergence check baseline is visible: as $i$ increases, the speed-up decreases while the solution quality increases. The EI and PI based stopping criteria behave similarly in terms of both RTC and RYC scores. The methods tend to stop BO very early, thus leading to significant speed up. However, maybe not surprisingly, such an aggressive early stopping leads to worse test performance on average. The convergence check baseline and our method could achieve various trade-offs between speed and solution quality effectively by changing patience hyperparameter or the post-correction term, and our method prioritizes quality over speed.

Furthermore, the standard deviations of the RYC and RTC scores also provide us with interesting insights. The RYC variances of our method are usually smaller than the baselines ones, indicating that we successfully maintain a high solution quality across a wide range of scenarios. On the other hand, the RTC variances of our method are usually higher than the baselines, which highlights that our method adapts to different scenarios rather than stopping BO at similar iterations.

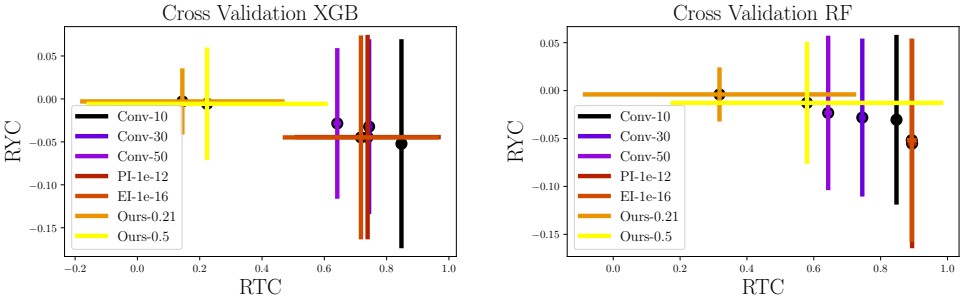

Figure 3: The mean and standard deviation of RYC and RTC scores for the compared automatic termination methods when using cross validation in the hyperparameter evaluation when tuning XGB (left) and RF (right). The mean value is shown as the large dot and the standard deviation is shown as an error bar in both dimensions.

## 4.4 NEURAL HYPERPARAMETER AND ARCHITECTURE SEARCH

A main advantage or our termination criterion is that it is applicable for any iterative HPO method. To demonstrate this, we apply it to several state-of-the-art Bayesian optimization methods from the literature: TPE (Bergstra et al., 2011), BORE Tiao et al. (2021), GP-BO (Snoek et al., 2012) as well as random search (RS) (Bergstra & Bengio, 2012). For each method and dataset, we perform 50 independent runs with a different seed.

We consider two popular tabular benchmark suites from the literature: NAS-HPO-Bench (Klein & Hutter, 2019), which mimics the hyperparameter and neural architecture search of multi-layer perceptrons on tabular regression datasets, and NAS-Bench-201 (Dong & Yang, 2020) for neural architecture search on image classification datasets. Notice that for NAS-Bench-201, we used *validation* metrics to compute RYC instead of test metrics, thus, no positive RYC scores are observed. For a detailed description of these benchmarks we refer to the original paper. We consider the following thresholds on the final regret $\{0.0001, 0.001, 0.01\}$ in our experiments, corresponding to a loss of performance of $0.01\%$, $0.1\%$ and $1\%$ compared to global optimum. Due to space constraints we will only show results for BORE in Figure Fig. 4 and Fig. 5, respectively, and provide all other results in Fig. 6 in the appendix.

While *no method* Pareto dominates the others, our termination criterion shows a similar trend as in Section 4.3 and tends to prioritise accuracy over speed. Users need to choose the threshold based on their own preference with respect to the speed-accuracy tradeoff, i.e a higher threshold saves more wall-clock time but potentially leads to a higher drop in performance. However, our criterion benefits from an intuitive and interpretable threshold. We further show a distribution of true regrets at the stopping iteration triggered by our method with the considered thresholds on HPO-Bench in Fig. 5d.

From Fig. 5d, with a high threshold of $0.01$, all the experiments (4 datasets with 50 replicates) are early stopped by our method and 41 (20%) experiments end up with true regret being higher than the threshold. With a low threshold of 0.0001, 112 experiments are stopped and 12 (10.7%) experiments end up with true regret being higher than the threshold. In short, our method achieves 80% to 90% success rate of stopping the BO with true regret within the user-defined tolerance.

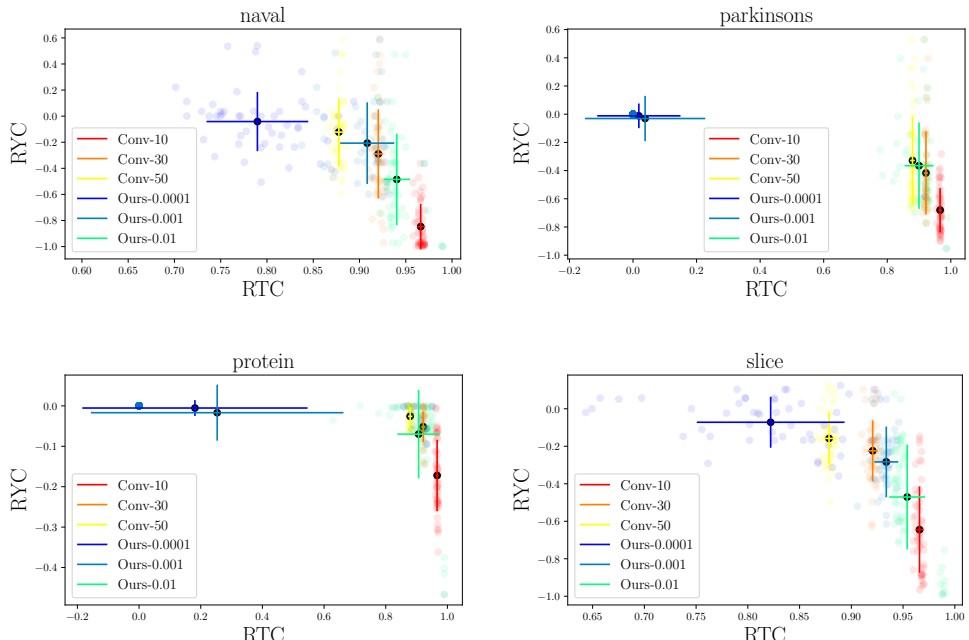

Figure 4: The mean and standard deviation of RYC and RTC scores for considered automatic termination methods for HPO-Bench datasets. The mean value is shown in the large dot and the standard deviation is shown as an error bar in both dimensions.

For every method we aggregated the scores over datasets with other HPO optimizers in Fig. 6. We can see that the speed up of the convergence check baseline is affected very mildly by the optimizers while the RYC scores largely depend on the optimiser: RYC scores with random search are worse than with BORE. In contrast, the RYC scores for our termination criterion are similar across optimisers, especially for smaller thresholds. On the other hand, the speed up for a given threshold tends to vary. This can be explained by the difference of the optimizer's performance, for examples random search is not as efficient as BORE, and hence the regret is mostly above the stopping threshold. In summary, while convergence check baselines are by design robust in terms of time saved, our method is more robust in terms of maintaining the solution quality.

## 4.5 OVERFITTING IN BO FOR HYPERPARAMETER OPTIMIZATION

Proposition 1 emphasises an important problem of BO-based HPO: while focusing (and minimizing) the validation error, we cannot fully reduce the discrepancy between the validation and test errors. Empirically, we show this might happen when correlation between the test and validation errors is low, thus improvement in validation performance does not lead to the better test results. A particular example of such low correlation in the *small* error region is presented in Fig. 8 (Appendix A.2) when tuning XGB and Random Forest on tst-census dataset.

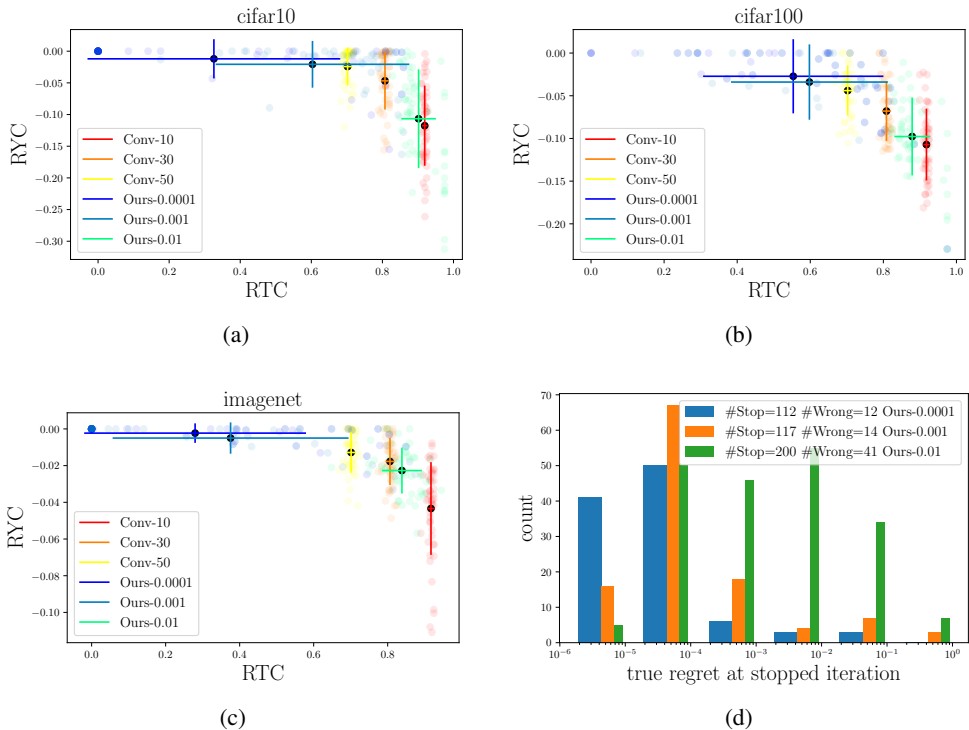

(a)

(b)

(c)

(d)

Figure 5: Fig. (a) - (c), the mean and standard deviation of RYC and RTC scores for considered automatic termination methods for NAS-Bench-201 datasets. Notice that *validation metrics* are used in these experiments, thus no positive RYC scores are observed. The mean value is shown in the big dot and the standard deviation is shown as error bar in both dimensions. In Fig. 5d, it shows a distribution of true regrets at the stopping iteration triggered by our method with considered thresholds for HPO-Bench. The number of stopped experiments and the number of "wrong" cases where the true regret is larger than the threshold are shown in the legend in Fig. 5d.

In our experiments, a positive RYC score is an indicator of overfitting, showing that the test error at the terminated iteration is lower than the test error in the final round. We observe positive RYC scores in both Fig. 3 and Fig. 4, one with cross-validation on small datasets and one with medium-sized datasets. Hence, we would like raise attention to the possible overfitting issue that occurs in HPO for which our method can be used as a plugin to mitigate overfitting.

## 5    CONCLUSION

Despite the usefulness of hyperparameter optimizations (HPO), setting a budget in advance remains a challenging problem. In this work, we propose an automatic termination criterion that can be plugged into many common HPO methods. The criterion uses an intuitive and interpretable upper bound of simple regret, allowing users explicitly control the accuracy loss. In addition, when cross validation is used in the evaluations of hyperparameters, we propose to use an analytical threshold rooted from the variance of cross validation results.

The experimental results suggest that our method can be robustly used across many HPO optimizers. Depending on the user-defined thresholds, with 80% to 90% chance, our method achieves true regret within that threshold, saving unnecessary computation and reducing energy consumption. We also observe that overfitting exists in HPO even when cross-validation is used. We hope our work will draw the attention of the HPO community to the practical questions of how to set budget in advance and how to mitigate overfitting when tuning hyperparameters in machine learning.

**Ethics Statement** In a broader context, we highlight that BO can reduce the computational cost required to tune ML models, mitigating the electricity consumption and carbon footprint associated with brute force techniques such as random and grid search. The automatic termination criterion we presented in this work can have a positive societal impact by further reducing the cost of tuning ML models. On the other hand, BO is a general methodology to optimize gradient-free functions and is not limited to specific application domains. Our early-stopping approach does not decrease the risk for misuse, calling for methods to enforce fairness constraints Perrone et al. (2021) as well as for care at model-deployment time.

**Reproducibility Statement** To improve the reproducibility, the code to rerun the experiments can be found at `https://anonymous.4open.science/r/BO-early-stopping-555E`, which will be made publicly available after the paper is published. Besides, we also detail the experiments setting in relevant aspects: the choices for BO are in Appendix A.1.1, the search spaces of the algorithms are in Appendix A.1.3 and the datasets sources and descriptions are in Appendix A.1.4.

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

# A APPENDIX

## A.1 EXPERIMENTS SETTING

### A.1.1 BO SETTING

We used an internal BO implementation where expected improvement (EI) together with Mat'ern-52 kernel in the GP are used. The hyperparameters of the GP includes output noise, a scalar mean value, bandwidths for every input dimension, 2 input warping parameters and a scalar covariance scale parameter. The closest open-source implementations are GPyOpt using input warped GP [1] or AutoGluon BayesOpt searcher [2]. We maximize type II likelihood to learn the GP hyperparameters in our experiments.

### A.1.2 ALGORITHM

### A.1.3 SEARCH SPACES FOR CROSS VALIDATION EXPERIMENTS

XGBoost (XGB) and RandomForest (RF) are based on scikit-learn implementations and their search spaces are listed in Table 1.

### A.1.4 DATASETS IN CROSS VALIDATION EXPERIMENTS

We list the datasets that are used in our experiments, as well as their characteristics and sources in Table 2. For each dataset, we first randomly draw 20% as test set and for the rest, we use 10-fold cross validations for regression datasets and 10-fold stratified cross validation for classification datasets. The actual data splits depend on the seed controlled in our experiments. For a given experiment, all the hyperparameters trainings use the same data splits for the whole tuning problem. For the experiments without cross-validation, we use 20% dataset as validation set and the rest as training set.

## A.2 DETAILED RESULTS

We first show the scatter plots of RTC and RYC scores for different automatic termination methods on HPO-Bench-datasets in Fig. 6 and the results on NAS-Bench-201 in Fig. 7.

### A.2.1 DETAILED NUMBERS OF RYC AND RTC SCORES

We report detailed RYC scores and RTC scores of different HPO automatic termination methods for the experiments in the main text in Table 3, Table 4 and Table 5.

---

[1] https://github.com/SheffieldML/GPyOpt
[2] https://github.com/awslabs/autogluon

---

**Algorithm 1** BO for HPO with cross-validation and automatic termination

---

**Require:** Model $\mathcal{M}_\gamma$ parametrized by $\gamma \in \Gamma$ , data $\{\mathcal{D}_1, \ldots, \mathcal{D}_k\}$ for $k$-fold cross-validation,
    acquisition function $\alpha(\gamma)$
1: Initialize $y_t^* = +\infty$ and $G_t = \{\}$
2: **for** $t = 1, 2, \ldots$ **do**
3:     Sample $\gamma_t \in \arg\max_{\gamma \in \Gamma} \alpha(\gamma)$
4:     **for** $i = 1, 2, \ldots, k$ **do**
5:         Fit the model $\mathcal{M}_\gamma(\cdot; \mathcal{D}_{-i})$, where $\mathcal{D}_{-i} = \cup_{j \neq i} \mathcal{D}_i$
6:         Evaluate the fitted model $y_t^i = \frac{1}{|\mathcal{D}_i|} \sum\limits_{x_i, y_i \in \mathcal{D}_i} \ell(y_i, \mathcal{M}_\gamma(x_i, \mathcal{D}_{-i}))$
7:     **end for**
8:     Calculate the sample mean $y_t = \frac{1}{k} \sum\limits_k y_t^i$,
9:     **if** $y_t \leq y_t^*$ **then**
10:        Update $y_t^* = y_t$ and the current best $\gamma_t^* = \gamma_t$
11:        Calculate the sample variance $s_{\text{cv}}^2 = \frac{1}{k} \sum_i (y_t - y_t^i)^2$
12:        Calculate the variance estimate $\text{Var}\hat{f}(\gamma_t^*) \approx \left( \frac{1}{k} + \frac{|\mathcal{D}_i|}{|\mathcal{D}_{-i}|} \right) s_{\text{cv}}^2$ from Eq. (8)
13:     **end if**
14:     Update $G_t = G_{t-1} \cup \gamma_t$ and $y_{1:t} = y_{1:t-1} \cup y_t$
15:     Update $\sigma_t, \mu_t$ with Eqs. (1) and (2)
16:     Calculate upper bound $\bar{r}_t := \min\limits_{\gamma \in G_t} \text{ucb}_t(\gamma) - \min\limits_{\gamma \in \Gamma} \text{lcb}_t(\gamma)$ for simple regret from Eq. (7)
17:     **if** the condition $\bar{r}_t \leq \sqrt{\text{Var}\hat{f}(\gamma_t^*)}$ holds **then**
18:        **terminate BO loop**
19:     **end if**
20: **end for**
21: **Output:** $\gamma_t^*$

---

Table 1: Search spaces description for each algorithm.

| tasks | hyperparameter | search space | scale |
|---|---|---|---|
| | n_estimators | $[2, 2^9]$ | log |
| | learning_rate | $[10^{-6}, 1]$ | log |
| | gamma | $[10^{-6}, 2^6]$ | log |
| | min_child_weight | $[10^{-6}, 2^5]$ | log |
| XGBoost | max_depth | $[2, 2^5]$ | log |
| | subsample | $[0.5, 1]$ | linear |
| | colsample_bytree | $[0.3, 1]$ | linear |
| | reg_lambda | $[10^{-6}, 2]$ | log |
| | reg_alpha | $[10^{-6}, 2]$ | log |
| | n_estimators | $[1, 2^8]$ | log |
| RandomForest | min_samples_split | $[0.01, 0.5]$ | log |
| | max_depth | $[1, 5]$ | log |

### A.2.2   CORRELATION BETWEEN VALIDATION AND TEST METRICS

In Fig. 8, we show the correlation between validation and test metrics of hyperparameters when tuning XGB and RF on tst-census dataset in Fig. 8.

### A.2.3   THE CHOICE OF $\beta_t$

High-probability concentration inequalities (aka confidence bounds) are important to reason about the unknown objective function and are used for theoretically grounded convergence guarantees in some (GP-UCB-based) BO methods (Srinivas et al., 2010; Ha et al., 2019; Kirschner et al., 2020; Makarova et al., 2021). There, $\beta_t$ stands for the parameter that balances between exploration vs. exploitation and ensures the validity of the confidence bounds. The choice of $\beta_t$ is then guided by the assumptions made on the unknown objective, for example, the

| dataset | problem_type | n_rows | n_cols | n_classes | source |
|---|---|---|---|---|---|
| openml14 | classification | 1999 | 76 | 10 | openml |
| openml20 | classification | 1999 | 240 | 10 | openml |
| tst-hate-crimes | classification | 2024 | 43 | 63 | data.gov |
| openml-9910 | classification | 3751 | 1776 | 2 | openml |
| farmads | classification | 4142 | 4 | 2 | uci |
| openml-3892 | classification | 4229 | 1617 | 2 | openml |
| sylvine | classification | 5124 | 21 | 2 | openml |
| op100-9952 | classification | 5404 | 5 | 2 | openml |
| openml28 | classification | 5619 | 64 | 10 | openml |
| philippine | classification | 5832 | 309 | 2 | data.gov |
| fabert | classification | 8237 | 801 | 2 | openml |
| openml32 | classification | 10991 | 16 | 10 | openml |
| openml34538 | regression | 1744 | 43 | - | openml |
| tst-census | regression | 2000 | 44 | - | data.gov |
| openml405 | regression | 4449 | 202 | - | openml |
| tmdb-movie-metadata | regression | 4809 | 22 | - | kaggle |
| openml503 | regression | 6573 | 14 | - | openml |
| openml558 | regression | 8191 | 32 | - | openml |
| openml308 | regression | 8191 | 32 | - | openml |

Table 2: Datasets used in our experiments including their characteristics and sources.

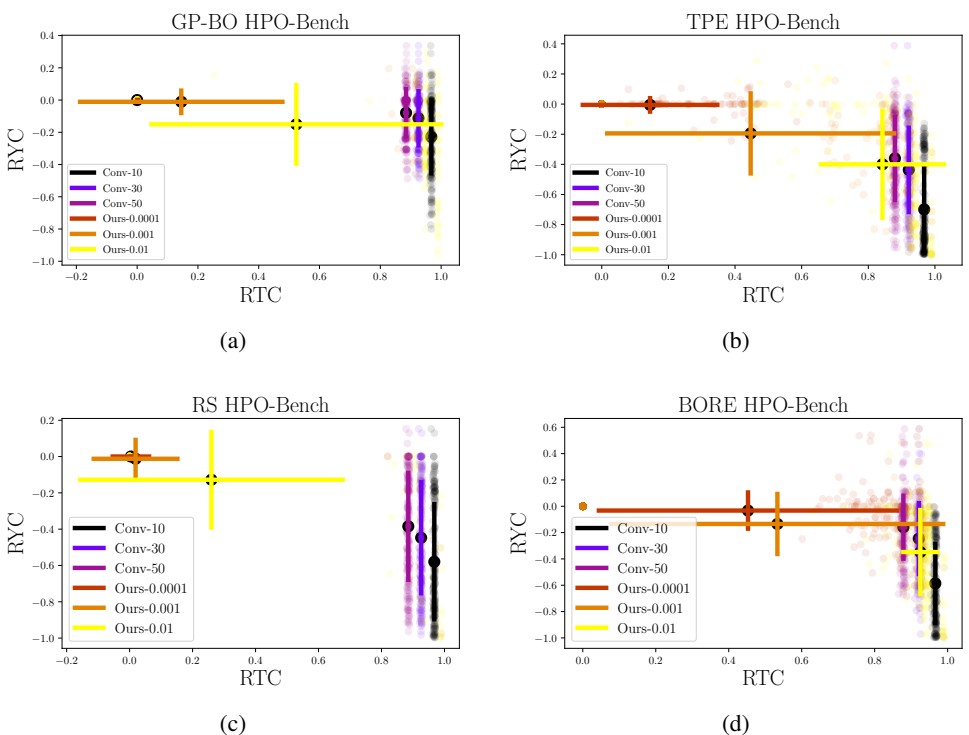

Figure 6: Fig. (a) - (d), the mean and standard deviation of RYC and RTC scores for considered automatic termination methods on HPO-Bench datasets using GP based BO (GP-BO), Random Search (RS), TPE and BORE optimizers. The mean value is shown in the big dot and the standard deviation is shown as error bar in both dimensions.

objective being a sample from a GP or the objective having the bounded norm in RKHS (more agnostic case used in Section 3).

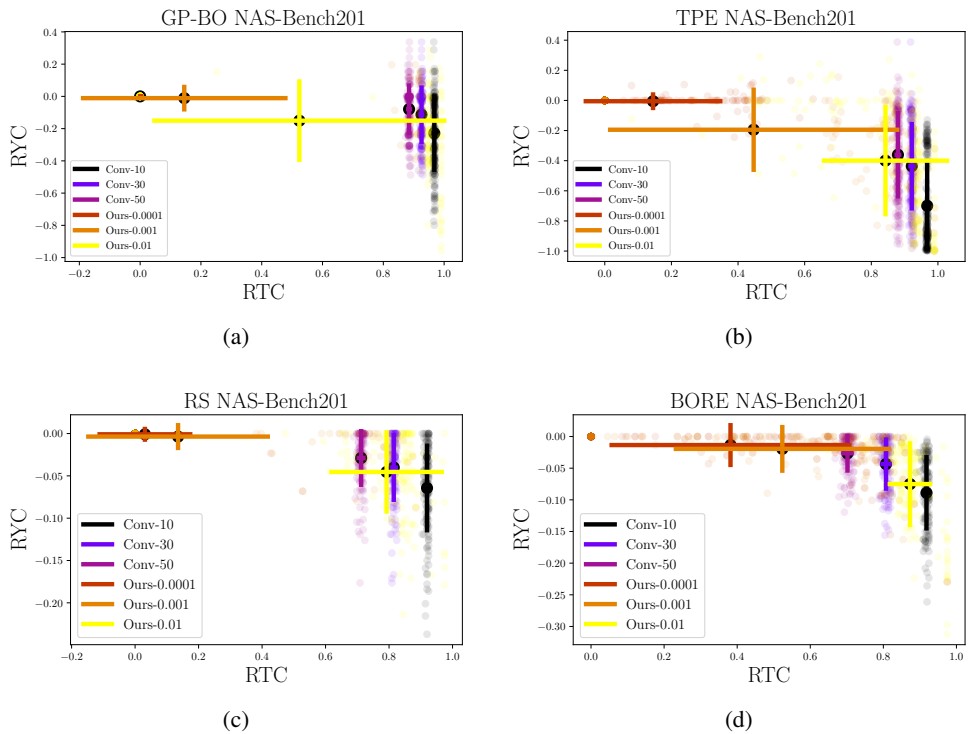

Figure 7: Fig. (a) - (d), the mean and standard deviation of RYC and RTC scores for considered automatic termination methods on NAS-Bench-201 datasets using GP based BO (GP-BO), Random Search (RS), TPE and BORE optimizers. The mean value is shown in the big dot and the standard deviation is shown as error bar in both dimensions.

| algo | RTC | | RYC | |
|---|---|---|---|---|
| | RF | XGB | RF | XGB |
| Conv_10 | 0.840 | 0.841 | -0.031 | -0.051 |
| Conv_30 | 0.686 | 0.666 | -0.022 | -0.026 |
| Conv_50 | 0.498 | 0.504 | -0.015 | -0.021 |
| EI_1e-08 | 0.896 | 0.850 | -0.057 | -0.052 |
| EI_1e-12 | 0.895 | 0.779 | -0.055 | -0.047 |
| EI_1e-16 | 0.893 | 0.718 | -0.052 | -0.045 |
| PI_0.0001 | 0.898 | 0.875 | -0.059 | -0.059 |
| PI_1e-08 | 0.895 | 0.814 | -0.055 | -0.052 |
| PI_1e-12 | 0.894 | 0.739 | -0.055 | -0.044 |
| Ours_0.21 | 0.318 | 0.144 | -0.004 | -0.003 |
| Ours_0.5 | 0.580 | 0.224 | -0.013 | -0.006 |

Table 3: RTC and RYC scores for early stopping methods in cross validation benchmarks.

In our experiments, we follow the common practice of scaling down $\beta_t$ which is usually used to improve performance over the (conservative) theoretically grounded values (see e.g., Srinivas et al. (2010); Kirschner et al. (2020); Makarova et al. (2021)). Particularly, throughout this paper, we set $\beta_t = 2 \log(|\Gamma| t^2 \pi^2 / 6\delta)$ where $\delta = 0.1$ and $|\Gamma|$ is set to be the number of hyperparameters. We then further scale it down by a factor of 5 as defined in the experiments in Srinivas et al. (2010). We provide an ablation study on the choice of $\beta_t$ in Fig. 9.

| | | RTC | | | | RYC | | |
| --- | --- | --- | --- | --- | --- | --- | --- | --- |
| dataset | naval | parkinsons | protein | slice | naval | parkinsons | protein | slice |
| Conv_10 | 0.943 | 0.947 | 0.946 | 0.942 | -0.605 | -0.582 | -0.117 | -0.432 |
| Conv_30 | 0.826 | 0.837 | 0.837 | 0.840 | -0.064 | -0.235 | -0.021 | -0.119 |
| Conv_50 | 0.748 | 0.729 | 0.734 | 0.747 | -0.038 | -0.107 | -0.008 | -0.058 |
| Ours_0.0001 | 0.790 | 0.018 | 0.198 | 0.822 | -0.041 | -0.012 | -0.005 | -0.072 |
| Ours_0.001 | 0.910 | 0.038 | 0.271 | 0.934 | -0.220 | -0.031 | -0.018 | -0.281 |
| Ours_0.01 | 0.941 | 0.901 | 0.906 | 0.953 | -0.498 | -0.378 | -0.071 | -0.466 |

Table 4: RTC and RYC scores for early stopping methods in HPO-Bench.

| | | RTC | | | RYC | |
| --- | --- | --- | --- | --- | --- | --- |
| dataset | ImageNet | cifar10 | cifar100 | ImageNet | cifar10 | cifar100 |
| Conv_10 | 0.880 | 0.889 | 0.888 | -0.034 | -0.098 | -0.097 |
| Conv_30 | 0.612 | 0.611 | 0.606 | -0.010 | -0.019 | -0.036 |
| Conv_50 | 0.372 | 0.361 | 0.372 | -0.004 | -0.006 | -0.014 |
| Ours_0.0001 | 0.274 | 0.311 | 0.519 | -0.002 | -0.008 | -0.026 |
| Ours_0.001 | 0.377 | 0.622 | 0.582 | -0.005 | -0.023 | -0.033 |
| Ours_0.01 | 0.837 | 0.902 | 0.879 | -0.022 | -0.106 | -0.099 |

Table 5: RTC and RYC scores for early stopping methods in NAS-Bench-201.

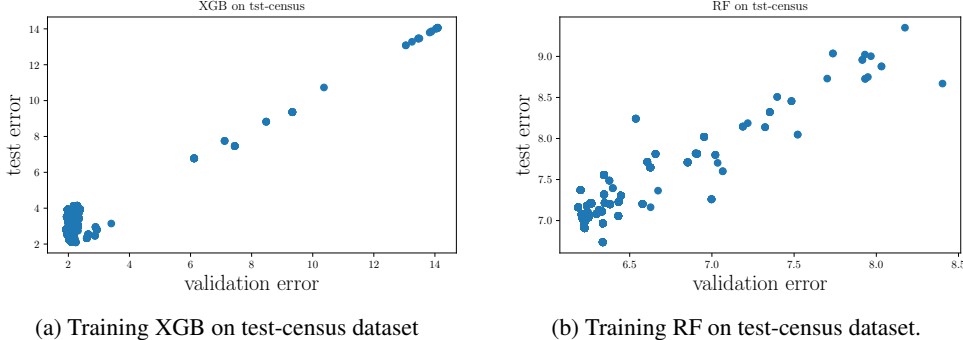

(a) Training XGB on test-census dataset

(b) Training RF on test-census dataset.

Figure 8: We show validation error for training XGB (a) and RF (b) on tst-census dataset on the $x$-axis and test error on the $y$-axis. In the *low* error region, the validation metrics are not well correlated with the test metrics.

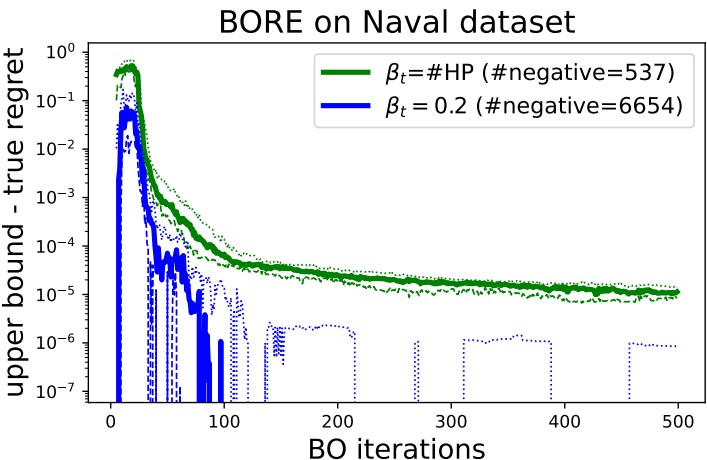

Figure 9: The differences between upper bound and true regret for every BO iterations when using BORE to tune an MLP on the Naval dataset. The number of negative differences (the upper bound is smaller than the true regret) are shown in the legend next to the two options for computing $\beta_t$.

