# OpenReview forum: "Automatic Termination for Hyperparameter Optimization"
_ICLR.cc/2022/Conference — ICLR 2022 Submitted_

### Official Review · Reviewer_MRmf · 2021-10-24

**Correctness:** 3
**Technical Novelty And Significance:** 3
**Empirical Novelty And Significance:** 3
**Recommendation:** 6
**Confidence:** 3

**Main Review:**

The problem setting of early terminating the entire BO process is interesting and novel. How to effectively pre-specify a termination criterion is a practical challenge in the field of HPO. The proposed mechanism of bounding the regret based on a user-defined tolerance makes intuitive sense. It is also reasonable to notice that overfitting can occur in HPO. Here are a few detailed comments:

- Figure 1a is not very visually informative. It is hard to understand that the orange line is the ucb and the green line is the lcb visually.
- Some typos: "prooved"-->"proved" in page 3, "letter"--> "latter" in page 4.
- It is hard to understand intuitively why the suboptimal solution resulted from the empirical surrogate can be mitigated by early stopping. It is suspicious to draw the conclusion that it can be mitigated by the proposed method. Could you explain more?

Quality: The submission is technically sound. The claims in the contribution are well-supported by theoretical analyses and empirically results. It is a complete piece of work that outperforms prior work empirically.

Clarity: This paper is well-written and easy to follow. The problem is also well-motivated. The experimental details are also very specific, such that reproducing the results should be possible.


**Summary Of The Paper:**

This paper studies the problem of pre-specifying the optimal termination criterion for Bayesian optimization. Different from prior work that tracks the value of acquisition function, this paper proposes an automatic termination criterion for BO. In particular, they construct a high-probability confidence bound on the regret, and then the users can specify a desired tolerance that shows how accurate the final solution should be compared to the global optimal. They estimate the threshold via a cross-validation estimate of the generalization error. Empirically, they design two evaluation metrics, relative test error change RYC and relative time change RTC, and compare to the comprehensive prior work. The results demonstrate the effectiveness of their proposed approach.


**Summary Of The Review:**

This paper proposes a novel early stopping criterion for Bayesian optimization. It is well-written and well-motivated. The proposed idea is simple and technically sound. The claims are well-supported by theoretical analyses and extensive experimental results.

---

> ### Author Response · Authors · 2021-11-19
> **Response to Reviewer MRmf**
>
> We thank the reviewer for the positive feedback and suggestions. We also think that early stopping the entire BO loop is a practically important problem and that bounding the regret is an effective way to achieve this. Not only does this spare users the need to pre-specify the number of BO evaluations, but it also saves them computational resources when continuing HPO is not expected to bring benefits.
>
> We incorporate edits in the revised paper and also answer the questions below.
>
> $\textbf{Q1:}$  Intuition on why the suboptimal solution resulted from the empirical surrogate can be mitigated by early stopping.
>
> $\textbf{A1:}$  We emphasize that our main goal is to prevent BO from consuming compute resources when not needed. Specifically, we argue that one should stop optimizing when the maximum plausible improvement drops below the standard deviation of the cross-validation estimator. We do not conclude that terminating BO early will provide a better-performing solution, but rather that terminating BO early allows you to save computational resources while not degrading the performance of the final solution. On one hand, this lifts the burden of specifying the number of BO iterations, which is hard to do a priori. On the other hand, this saves computational resources that would not lead to a better performance quality for the detected hyperparameter configuration.
>
>
> $\textbf{Q2:}$   Quality: The submission is technically sound. The claims in the contribution are well-supported by theoretical analyses and empirically results. It is a complete piece of work that outperforms prior work empirically.
> Clarity: This paper is well-written and easy to follow. The problem is also well-motivated. The experimental details are also very specific, such that reproducing the results should be possible.
>
> $\textbf{A2:}$ Thank you for the positive feedback. We hope that our response has addressed your questions and that you will consider increasing your score.

---

### Official Review · Reviewer_QxQD · 2021-10-27

**Correctness:** 3
**Technical Novelty And Significance:** 2
**Empirical Novelty And Significance:** 2
**Recommendation:** 5
**Confidence:** 4

**Details Of Ethics Concerns:**

N.A.

**Main Review:**

Lack of literature review:
In general, this paper aims to solve the early/optimal stopping of the Bayesian optimization (BO). Plenty of related works are even not mentioned in the paper, e.g. Freeze-thaw BO [1],  Multi-fidelity BO [2], Hyperband BO [3], and BOS-BO [4], these works all discussed how to achieve efficiency and saving budget in BO. A detailed discussion of the difference between the proposed method and all the previous works needs to be mentioned, not to mention comparison experiments.

The stopping criterion is mainly decided by Eq. (7), specifically, determined by the upper confidence bound and lower confidence bound. However, this bound is quite loose, since there is one variable determining the magnitude of it: $\beta_t$. Through the paper, the authors do not mention the role of $\beta_t$ and its impact on regret. In fact, if $\beta$ is set small enough, the condition in Proposition 2 can be easily satisfied right?

Moreover, the variance estimate of $\hat{f}$ lacks theoretical analysis. I am expecting the authors to giving theoretical guarantee of the estimate to the real variance. Otherwise, it won't be possible to determine whether the estimate is to the same magnitude of the real variance.


[1] Swersky, K., Snoek, J., and Adams, R. P. Freeze-thaw Bayesian optimization. arXiv:1406.3896, 2014.

[2] Kandasamy, K., Dasarathy, G., Oliva, J. B., Schneider, J., and Poczos, B. Gaussian process bandit optimization with multi-fidelity evaluations. In Proc. NIPS, 2016.

[3] Falkner, S., Klein, A., and Hutter, F. BOHB: Robust and efficient hyperparameter optimization at scale. In Proc. ICML, 2018.

[4] Dai, Z., et al. Bayesian optimization meets Bayesian optimal stopping. In Proc. ICML, 2019.

**Summary Of The Paper:**

This paper proposes an automatic termination criterion for Bayesian optimization (BO) by using the upper bound of the simple reget. Various experiments are conducted to demonstrate that with the utilization of the termination criterion, computation and energy consumption can be reduced. The major contribution of this paper lies in the two propositions. Proposition 1 discusses the relationship between statistical error and optimization error, the authors claim that due to the irreducible statistical error, it is appropriate to reduce the optimization error $\epsilon_{BO}$ to the same magnitude w.r.t. the statistical error. Since the statistical error $\epsilon_{st}$ is unknown, the authors adopt an existing cross-validate method to estimate it. Then in proposition 2, the termination criterion is detailed once the BO regret is less than the standard deviation of statistical variance.

**Summary Of The Review:**

Given the lack of literature review and comparison experiments, plus the bound for $\epsilon_{BO}$ lacks theoretical guarantee. This paper needs a major revision.

---

> ### Author Response · Authors · 2021-11-19
> **Response to Reviewer QxQD**
>
> We want to thank the reviewer for the insightful comments.  Below, we clarify the raised question.
>
> $\textbf{Q1:}$  Lack of literature review.
>
> $\textbf{A1:}$  The papers referenced by the reviewer can indeed save compute resources in BO, but in an $\textit{orthogonal way to our work}$ in that each iterative evaluation for a hyperparameter configuration (e.g., based on SGD) can be early stopped. In contrast, we aim to stop the $\textit{whole BO process}$, which is a different problem and an overlooked one in the HPO community. In practice, one could simply combine the two to achieve overall larger speed ups.
>
> $\textbf{Q2:}$  Role of $\beta_t$ and its impact on regret.
>
> $\textbf{A2:}$  In the revised version, we discuss in more detail $\beta_t$ in Appendix A.2.3 along with the ablation study in Figure 9 (in addition to the initial mention of its computation at the end of the second paragraph in Section 4.3). Particularly, in Appendix A.2.3, we refer to the theoretical derivations for $\beta_t$ (under different assumptions of the objective) as well as the common practice to choose $\beta_t$ used to improve performance over the (conservative) theoretically grounded values.
>
>
> $\textbf{Q3:}$ Theoretical analysis of the variance estimate of $\hat f$.
>
> $\textbf{A3:}$  It is true, we do not give the guarantee of the estimate to the real variance since we use the (theoretically grounded) results from Nadeau \& Bengio (2003). As these results are not the contribution of our work, we refer the reviewers (and readers) to the original paper for detailed analysis.

---

> > ### Author Response · Authors · 2021-11-22
> > **Further response to Reviewer QxQD**
> >
> > We would like to ask the reviewer whether our explanation clarifies the reviewer’s concerns.
> > If it is not the case, we are ready to provide further clarifications, if necessary.

---

> > > ### Author Response · Authors · 2021-11-28
> > > **Further response to Reviewer QxQD**
> > >
> > > We would like to draw the reviewer's attention, that a similar question regarding the variance (Q3) was additionally addressed in the discussion with the reviewer yUir where we provided more details. We hope that this clarifies the reviewer's concern. If it is not the case, we are ready to answer further (possible) questions while it is still possible.

---

### Official Review · Reviewer_yUir · 2021-10-31

**Correctness:** 2
**Technical Novelty And Significance:** 3
**Empirical Novelty And Significance:** 3
**Recommendation:** 6
**Confidence:** 4

**Main Review:**

Overall, I think the main idea of the work is interesting. The writing of the paper is generally clear, except Section 3, which I think is not mathematically rigorous. This section causes me a lot of confusion and I find it's hard to understand all the maths behind the proposed approach. I have several questions listed below:
1.	I'm confused by the definition of \epsilon_{BO}. Is \epsilon_{BO} a user-defined threshold or is there any formula showing the definition of \epsilon_{BO}? I can't find the definition of \epsilon_{BO} in the paper. Also, in Proposition 1, it says that \hat{\gamma}_D is a candidate solution of min_{\gamma \in \Gamma} \hat{f}(\gamma) such that \hat{f}(\hat{gamma}_D) - \hat{f}(\gamma^*_D) \leq \epsilon_{BO}, this means the set of \hat{\gamma}_D depends on \epsilon_{BO}. If this is the case, then I expect the notation of \hat{gamma}_D should be changed to reflect the dependent (because for different values of \epsilon_BO, there will be a different set of "candidate solutions".
2.	In the first sentence right after the proof of Proposition 1, it is stated that the optimization error \epsilon_{BO} is bounded by the simple regret \hat{r}_t? Why is this the case? If \epsilon_{BO} is a user-defined threshold then it shouldn’t have any relation with the simple regret? Why is it bounded by the simple regret? I think the confusion here is also related to my confusion on the definition of \epsilon_{BO}.
3.	In the paragraph "Upper bound for \hat{r}_t" at the end of Page 3, there is a statement saying that "with high probability, \hat{f}(\gamma) is bounded by lower and upper confidence bounds defined as lcb_t(\gamma) = \mu_t(\gamma) - \beta_t \sigma_t(\gamma) and ucb_t(\gamma) = \mu_t(\gamma) + \beta_t \sigma_t(\gamma)". This statement is not always true, it is only true when the domain of the objective function is discrete. When the domain is continuous, it needs to have another term O(1/t^2) in the upper and lower bound because of the discretization process (see Lemmas 5.5, 5.6, 5.7 in Srinivas et al (2010)).
4.	In Eq. (7), I do not understand why \epsilon_{BO} is smaller than \hat{r}_t. Can the authors explain in more detail?
5.	In the sentence right below Eq. (7), it states that \hat{f}(\gamma_t^*) = min_{\gamma \in G_t} \hat{f}(\gamma). On the other hand, In Eq. (6), it states that \hat{f}(\gamma_t^*) is the best value found by iteration t. Combining these two equations, this means \gamma_t^* is the value with the lowest value of \hat{f}(\gamma)? In practice, this is only true when the empirical loss function \hat{f}(\gamma) is noiseless. However, note that to apply the regret analysis in Srinivas et al (2010) the objective function needs to be noisy.
6.	I'm also confused with Proposition 2. Is this a proposal from the paper, that is, to terminate BO when \bar{r}_t < \sqrt{Var \hat{f} (\gamma_t^*)}? Or is this a mathematical proposition (i.e. when the condition in Eq. (9) occurs, BO is automatically terminated)? I can't find a proof of Proposition 2 in the Appendix. Also, by checking the Algorithm 1 in Page 12 in the Appendix, I have a feeling that this Proposition 2 is just a proposal from the paper. Is this the case?
7.	I found it's a bit hard to read Figure 3. More detailed explanation for Figure 3 is needed in the revised version of the paper.


**Summary Of The Paper:**

The paper proposes a new approach for automatic termination of hyperparameter optimization based on Bayesian Optimization (BO).  The idea is to construct high-probability confidence bound on the regret and then determine when to terminate the BO process. Empirical experiments are conducted on various real-world HPO and NAS benchmarks to show the efficacy of the proposed approach.

**Summary Of The Review:**

Like I mentioned in the previous section, I think the overall idea of the work is interesting. But the maths behind the proposed approach is not rigorous and confused. I find a lot of confusion reading Section 3, which is the main section of the paper. I have several questions to the authors listed in the previous sections, hopefully, the answers from the authors will be able to clear my doubt.

---

> ### Author Response · Authors · 2021-11-19
> **Response to Reviewer yUir**
>
> We would like to thank the reviewer for constructive feedback. We address the raised questions below.
>
>  $\textbf{Q1:}$  Definition of $\epsilon_{BO}$ and its usage along with $\hat r_t$ in the paper.
>
>  $\textbf{A1:}$ We have revised the usage of the simple regret $\hat r_t$ and $\epsilon_{BO}$ that led to some presentation changes in Section 3.1 (see the new paper version).  We clarified the difference between the simple regret $\hat r_t$, the value that is iteratively changed (and appears in Proposition 1), and $\epsilon_{BO}$, a fixed threshold (user-defined or automatically computed) on the regret for BO to be terminated. Then (answering your question on the relation), the simple regret $\hat r_t$ (or rather its estimate Eq.7) is compared to $\epsilon_{BO}$ at each iteration $t$, and BO is terminated if the regret falls below $\epsilon_{BO}$, i.e. at some iteration $T : \hat r_T \leq \epsilon_{BO}$. It is, however, an open question $\textit{how}$ to set this threshold $\epsilon_{BO}$,  that we then address in Section 3.2, Termination threshold, where we (motivated by Proposition 1) derive $\epsilon_{BO}$ in case of cross-validation (as an alternative to a user-defined threshold). We hope that clarifies the confusion.
>
>
>  $\textbf{Q2:}$  Notation of a candidate solution in Proposition 1 and its dependence on $\epsilon_{BO}$.
>
>  $\textbf{A2:}$  Given the notation changes mentioned above, the revised Proposition 1 operates with the simple regret $\hat r_t=\hat f(\gamma^*_t) - \hat f(\gamma^*_D)$ with $\gamma_t^*$ denoting the candidate solution for
> $\hat f(\gamma)$ minimization. We hope that clarifies the notation confusion, since now the $\gamma^*_t$-notation for the solution is consistent with $t$-dependent $\hat r_t$ along with the desired discrepancy $f(\gamma^*_t) - f(\gamma^*)$.
>
>
>  $\textbf{Q3:}$  Confidence bounds $lcb_t(\gamma)$ and $ucb_t(\gamma)$ (page 3) are only valid for a discrete domain and an extra-term $O(1/t^2)$ is required for a continuous domain.
>
>  $\textbf{A3:}$  The confidence bounds along with the $\beta_t$ indeed depend on the assumptions made on the optimized objective (and the domain). Particularly, for a function $f$ with bounded norm in the RKHS associated with the covariance function $k$ (as we assume at page 3), these confidence bounds differ from the ones for $f$ assumed to be a GP sample. More precisely, in [1], this difference is reflected in Theorem 6 for RKHS (our assumption; no dependence on $O(1/t^2)$) and Lemmas 5.5-5.7 for a GP sample. We stick to RKHS assumption since it is more general and widely used (along with GP prior being \textit{just} a modelling choice). Note, however, that the theoretically grounded $\beta_t$ (e.g., derived in Theorem 6 in [1]) are quite conservative, and it is a common practice to adapt $\beta_t$ in the experiments (e.g., in [1] $\beta_t$ is scaled by a factor of 5, $\beta_t=2$ in [2,3], $\beta_t=1$ in [4]).  In the revised version, we have added the discussion of this practice-theory gap for $\beta_t$ in Appendix A.2.3 along with the ablation study.
>
>
>  $\textbf{Q4:}$  Typo in In Eq. (7).
>
>  $\textbf{A4:}$  We apologize for the typo that was accidentally placed in the inequality. The point of Eq. (7) is that $\bar r_t$ upper bounds $\hat f(\gamma^*_t) - \hat f(\gamma^*_D)$, and in the revised version we updated Eq. (7) to avoid any confusion.
>
>
>  $\textbf{Q5:}$  Definition of $\hat f(\gamma_t^*)$ in Eq. (7) and (6), it's usage in simple regret and noiseless or noisy experiment setting.
>
> $\textbf{A5:}$  The correct way is to state $\hat f(\gamma_t^*) = \min_{\gamma \in G_t} \hat f(\gamma)$ when used in the regret according to the widely used definition (i.e., we corrected Eq. 6). In practice, the regret is indeed impossible to compute (function is unknown), that is why we derive the upperbounds for it in Eq. 7 that take $\textit{noise-perturbed evaluations}$ into account (see first equation Section 2 and cross-validation description Page 4).
>
>
>  $\textbf{Q6:}$  Proposition 2 -- a mathematical proposition or a proposal?
>
> $\textbf{A6:}$  Yes. In the revised version, we rephrased the proposition to make it clear that this is our proposal.
>
>
>  $\textbf{Q7:}$  Better explanation of Figure 3.
>
> $\textbf{A7:}$  We added more explanations for Figure 3 in Section 4.3.
>
> [1] N. Srinivas, A. Krause, S. Kakade, and M. Seeger. Gaussian process optimization in the bandit
> setting: No regret and experimental design. ICML 2010.
>
> [2] J. Kirschner, I. Bogunovic, S. Jegelka, and A. Krause. Distributionally robust Bayesian
> optimization. AISTATS 2020.
>
> [3] A. Makarova, I. Usmanova, I. Bogunovic and A. Krause. Risk-averse Heteroscedastic Bayesian Optimization. NeurIPS 2021.
>
> [4] S. Curi, I. Bogunovic and A. Krause. Combining Pessimism with Optimism for Robust and Efficient Model-Based Deep Reinforcement Learning. ICML 2021.

---

> > ### Comment · Reviewer_yUir · 2021-11-21
> > **Respond to the authors' response**
> >
> > I'd like to thank the authors for their response. The response does clear my confusion, and I think Section 3 is now much clearer and the proposed approach is now easier to understand. I decided to increase my score to 6 as I think the paper has merit and can be of interest to the community. But I can't increase to a higher score as I also agree with the other reviewer that the paper will be stronger when a rigorous theoretical analysis of the variance estimate of \hat{f} is conducted.

---

> > > ### Author Response · Authors · 2021-11-22
> > > **Further response to Reviewer yUir**
> > >
> > > We appreciate the reviewer’s response and the corresponding score update.
> > >
> > > We further (sincerely) believe that the query of “rigorous theoretical analysis of the variance estimate of \hat{f}” might have appeared due to misunderstanding. We would like to emphasize, that such an analysis $\textit{already}$ exists in [1] (Section 3). Particularly, it is shown how based on the unbiased estimator of the variance (see Eq. (7)) that contains unknown values, one can get the approximation.  For this approximation (used in our paper), the authors state when and why it can overestimate/underestimate the variance.
> > >
> > > Some further theoretical analysis of variance of cross-validation estimator can be found, for example, in [2], i.e. the whole paper dedicated to the question. We, thereby, want to note, that the question of variance analysis falls far beyond the Termination condition for BO and is itself a hard problem addressed by the works we refer to.
> > >
> > > To conclude, based on the feedback, we are happy to add an additional discussion to the paper summarising the ideas from [1, 2] (both cited already). In this case, any reader will be aware of the problem difficulty.
> > >
> > > We hope that explains the situation and would not further affect the reviewer’s decision regarding the paper.
> > >
> > > [1] Claude Nadeau and Yoshua Bengio. Inference for the generalization error. In Neural Information Processing Systems (NeurIPS), 2003.
> > >
> > > [2] Pierre Bayle, Alexandre Bayle, Lucas Janson, and Lester Mackey. Cross-validation confidence intervals for test error. In Neural Information Processing Systems (NeurIPS), 2020.

---

> > > > ### Author Response · Authors · 2021-11-28
> > > > **Further Response to Reviewer yUir**
> > > >
> > > > We would like to ask whether our explanation clarifies the reviewer’s concern about the theoretical analysis of the variance estimate. If it is not the case, we are ready to provide more careful details since we believe that a misunderstanding might have occurred.

---

> > > > > ### Comment · Reviewer_yUir · 2021-11-29
> > > > > **Thank you for the further response**
> > > > >
> > > > > Dear authors,
> > > > >
> > > > > Sorry for the late reply, and also thank you for the further response regarding the “rigorous theoretical analysis of the variance estimate of \hat{f}”. From my side, I don't think it's a misunderstanding, but it's indeed a requirement from me in order to rate the paper higher.
> > > > >
> > > > > I understand that some theoretical analyses of the variance of the cross-validation estimators can be found in other existing works, but in my opinion, the authors should incorporate those analyses into the analyses of the Bayesian optimization with automatic termination. The suggested termination condition for BO in Section 3.2 and all the explanations in this Section are reasonable, but not that mathematically rigorous. To me, it will be better when there are some rigorous analyses showing the property of the proposed BO approach (e.g. regret or something similar) when the termination condition occurs.
> > > > >
> > > > > Because of the lack of this analysis, so with the current contribution of the paper, I can only give a score of 6.

---

> > > > > > ### Author Response · Authors · 2021-11-30
> > > > > > **Further response to Reviewer yUir**
> > > > > >
> > > > > > We appreciate the reviewer’s response, but would kindly ask them to elaborate on their request.
> > > > > > Currently, the (simple) regret bound, as asked for by the reviewer, already appears in Eq. 9, i.e., when the termination threshold is reached, the regret is smaller than the estimation of the statistical error.  The typical cumulative regret bounds (i.e., R_T defined as a sum over simple regrets) arguably are not a good fit for studying termination criteria like ours, since they are defined w.r.t. a fixed (not variable) horizon T.  Therefore, we would really appreciate if the reviewer could elaborate on what specific type of analysis they consider missing.

---

### Decision · Program_Chairs · 2022-01-20

**Decision:**

Reject

**Comment:**

In this paper, the stopping condition of Bayesian Optimization (BO) is discussed. This problem is very important when BO is applied to the Hyper-parameter optimization (HPO) task. All the reviewers agree that the proposed approach based on high-probability confidence bound on the regret is interesting and reasonable.  An important issue raised by a reviewer is that many existing BO works discussed how to achieve efficiency and saving budget in BO although they did not explicitly mention the stopping condition. Due to the lack of discussion regarding the relationship with these highly related studies, we have to conclude that the paper cannot be accepted in its current form.